# Boron-Doped Diamond Electrodes for Toxins Sensing in Environmental Samples—A Review

**DOI:** 10.3390/s25072339

**Published:** 2025-04-07

**Authors:** Aleksandar Mijajlović, Vesna Stanković, Tijana Mutić, Sladjana Djurdjić, Filip Vlahović, Dalibor Stanković

**Affiliations:** 1Department of Analytical Chemistry, Faculty of Chemistry, University of Belgrade, Studentski trg 12-16, 11000 Belgrade, Serbia; dh022024@student.chem.bg.ac.rs (A.M.);; 2Department of Chemistry, Institute of Chemistry, Technology and Metallurgy, University of Belgrade, Njegoševa 12, 11000 Belgrade, Serbia; vesna.stankovic@ihtm.bg.ac.rs (V.S.); tijana.mutic@ihtm.bg.ac.rs (T.M.); filip.vlahovic@ihtm.bg.ac.rs (F.V.)

**Keywords:** boron-doped diamond, pesticides, heavy metals, pharmaceuticals, industrial pollutants

## Abstract

Boron-doped diamond electrodes have found applications in the detection, monitoring, and mitigation of toxic chemicals resulting from various industries and human activities. The boron-doped diamond electrode is a widely applicable technology in this field, primarily due to its excellent surface characteristics: minimal to no adsorption, a wide operating potential range, robustness, and high selectivity. These extraordinary properties can be further enhanced through surface termination, which can additionally improve the analytical performance of boron-doped diamond (BDD) electrodes. The high accuracy and precision of the developed methods indicate the broad practical applicability of these electrodes across various sample matrices. Some studies have shown that different strategies can lead to enhanced sensitivity and selectivity, such as modifying the electrode surface (nanostructuring), forming different composite materials based on BDD, or implementing miniaturization techniques. Thus, this review summarizes the recent literature on the electroanalytical applications of BDDE surfaces, with a particular focus on environmental applications.

## 1. Introduction

Massive amounts of hazardous chemicals, such as pesticides, heavy metals, and industrial pollutants, are discharged into the environment every year as a result of modern society’s fast industrialization and agricultural growth. These dangerous compounds pose serious risks to ecosystems and human health when they leak into groundwater, surface water, and even sources of drinking water. Many of these pollutants are highly hazardous and persistent, even at minimal levels, making it very difficult for natural mechanisms to break them down. The addition of heavy metals like lead, mercury, and cadmium, as well as industrial pollutants like phenols, dyes, and hazardous pesticides, adds to the complexity of wastewater treatment. These contaminants not only affect aquatic habitats but also accumulate in the food chain, posing long-term environmental and health risks. When left untreated or inadequately managed, they can have serious effects on biodiversity and societal sustainability [1,2,3].

Conventional methods for the detection of toxic chemicals, including pesticides, industrial pollutants, and heavy metals, have historically relied on advanced techniques such as gas chromatography (GC), liquid chromatography (LC), mass spectrometry (MS), and atomic absorption spectroscopy (AAS). While these approaches are highly effective for identifying and quantifying contaminants in various matrices, they face several critical limitations. For instance, these approaches frequently need sophisticated, labor-intensive, and time-consuming sample preparation procedures, which can limit their efficiency in high-throughput applications. The necessary instrumentation is expensive and requires specialist training to operate, making these technologies less accessible for everyday environmental monitoring, especially in resource-limited areas. Furthermore, these traditional approaches are often limited to laboratory settings, limiting their utility for in situ and real-time detection of pollutants in water, soil, and air. Given the growing global concern about the presence of these toxic substances and their impact on human health and ecosystems, there is an urgent need for innovative, cost-effective, and portable detection technologies. These advanced methods should not only provide high sensitivity and specificity but also enable real-time monitoring to address the growing environmental challenges posed by these hazardous pollutants [4,5,6].

Electrochemical sensors offer significant advantages over traditional methods for detecting toxic substances like pesticides, industrial pollutants, and heavy metals. They are cost-effective, portable, and easy to use, making them ideal for on-site applications and fieldwork. Their ability to deliver rapid results, combined with compact and portable equipment, makes them a practical choice for real-time monitoring of contaminants in water, soil, and air. Materials commonly used in electrochemical sensors include metals such as platinum and gold, along with carbon-based materials like glassy carbon, carbon nanotubes, and graphene, all valued for their excellent conductivity and chemical stability. However, working with complex environmental samples poses challenges, particularly due to electrode fouling. This occurs when target compounds, byproducts of reactions, or other impurities from the sample matrix accumulate on the electrode surface, decreasing the sensor’s sensitivity, accuracy, and durability during extended use [7,8].

To overcome these limitations, improvements in electrode materials and surface treatments have played an important role. One common approach involves electrochemical pretreatment, such as anodic or cathodic polarization, which changes the surface of the diamond electrode to either oxygen- or hydrogen-terminated. This affects how the surface interacts with water and how easily electrons are transferred. In addition, the electrode surface can be modified with nanomaterials, polymers, or recognition elements to improve selectivity and sensitivity. These modifications help adapt the electrode to specific target substances and lead to better performance, especially when analyzing complex environmental or biological samples. For instance, boron-doped diamond electrodes stand out due to their wide potential window, remarkable resistance to fouling, and robust durability. These innovations enhance the effectiveness of electrochemical sensors, enabling them to achieve ultra-low detection limits, detect multiple contaminants simultaneously, and maintain high selectivity in complex environmental conditions. As a result, these sensors are becoming invaluable tools for reliable and autonomous environmental monitoring [9,10].

BDD thin films, as a modern type of electrode material, have attracted considerable research interest in recent years. They are usually produced by chemical vapor deposition (CVD), a technique that allows precise control over film thickness, crystal structure, and the amount of boron added. The electrochemical properties of these films largely depend on the level of boron doping—higher levels increase conductivity while maintaining the wide potential window typical of diamond electrodes. Their nanocrystalline or microcrystalline structure provides a large active surface area, which can help improve sensitivity in sensor applications. In addition, BDD films show strong long-term stability and resistance to chemical damage, making them well-suited for use in reliable electrochemical sensors for environmental and biological monitoring [11,12].

Thanks to their exceptional properties, BDDE is widely used for detecting toxic substances such as pesticides, industrial pollutants, and heavy metals. The effectiveness of electrochemical detection relies heavily on the interaction between the electrode surface and the analyte within the electrolyte solution. Therefore, the surface structure and characteristics of BDDEs, including nanostructured and chemically modified surfaces, are crucial to their overall performance [13].

Compared to traditional carbon-based electrodes like glassy carbon and carbon paste, BDD electrodes have several important advantages. These include a very wide potential window, very low background current, strong mechanical and chemical stability, and high resistance to surface contamination. While carbon electrodes can have a narrow working range and may degrade over time, BDD electrodes remain stable even in harsh chemical environments and complex sample types. Because of these features, BDD electrodes are well-suited for long-term use and are especially effective in detecting pollutants [11].

Several studies published in recent years highlight the growing interest in this field. According to Scopus, the majority of research papers were published between 2017 and 2025, with a notable peak in 2020, when approximately 80 research papers focused on the application of BDDEs (around 300 documents) in various fields of chemistry were published (Figure 1).

This paper highlights the application of BDDEs in detecting toxic chemicals in various environmental matrices.

## 2. Theoretical Insights into BDDE Applications in Toxic Chemical Sensors

Boron-doped diamond electrode (BDDE) has emerged as an adaptable and high-performance material for electrochemical applications. Their distinctive features result from the incorporation of boron into the diamond lattice, which converts the otherwise insulating diamond into a conductive substance. This doping creates midgap states, which greatly improve electrical conductivity and electron transfer reactivity. BDDE has a large electrochemical potential window, which allows for the detection and analysis of analytes that would otherwise interfere with oxygen and hydrogen evolution reactions. This characteristic makes them suitable for analyzing species in aqueous solutions without the limitations imposed by traditional electrodes. Diamond’s crystalline structure with sp^3^ hybridization ensures chemical inertness and mechanical durability, making it suitable for usage in harsh situations. Furthermore, the low background current observed with BDDE contributes to their great sensitivity, making them suitable for detecting trace quantities of analytes such as heavy metals, pesticides, and industrial pollutants [14].

Electrochemical reactions on BDDEs take place at the interface between the electrode surface and the electrolyte. The surface properties play a crucial role in their performance, with hydrogen-terminated surfaces being hydrophobic and oxygen-terminated surfaces exhibiting hydrophilic behavior. Moreover, the lack of surface oxides and their resistance to fouling contribute to the exceptional stability and durability of BDDEs. These qualities, along with their ability to be functionalized, make BDDEs a highly effective choice for environmental monitoring and analytical applications [15].

Compared to traditional electrodes such as glassy carbon, gold, and platinum, BDDEs have a number of benefits. Their remarkably broad electrochemical potential window, which enables the detection of analytes that are often obscured by water oxidation or reduction events on conventional electrodes, is one of their most noteworthy advantages. Furthermore, BDDEs have a substantially lower background current than metal and glassy carbon electrodes, which improves sensitivity and permits trace-level detection.

BDDEs resist biofouling and organic contamination more effectively than glassy carbon, ensuring reliable and reproducible measurements. These characteristics position BDDEs as superior alternatives for demanding analytical and environmental monitoring tasks (Figure 2) [16,17].

BDDEs are utilized in a variety of electrochemical techniques, such as anodic and cathodic voltammetry (ASV and CSV), differential pulse voltammetry (DPV), and cyclic voltammetry (CV). Anodic voltammetry is ideal for detecting substances that undergo oxidation at positive potentials, while cathodic voltammetry targets reduction reactions. DPV stands out for its ability to enhance both sensitivity and selectivity by reducing background noise and emphasizing specific redox processes. When paired with the unique properties of BDDEs, these techniques allow for accurate, reliable, and adaptable detection of analytes, even in complex environmental matrices [18,19].

## 3. Detecting Heavy Metals: Advanced Detection Strategies for a Safer Environment

Heavy metals in the environment are a major danger because of their toxicity, persistence, and potential to bioaccumulate in living species. Elements like lead (Pb), mercury (Hg), and cadmium (Cd) are especially dangerous because they interfere with biological functions even at low levels. Heavy metals, once discharged into the environment, can last for decades, cycling via soil, water, and air before entering the food chain [20].

Bioaccumulation of heavy metals in organisms causes elevated concentrations at higher trophic levels, threatening human health and biodiversity. Long-term exposure to these elements has been related to neurological abnormalities, organ damage, developmental problems, and carcinogenic effects. Mercury, for example, easily converts into methylmercury in aquatic systems, where it accumulates in fish, providing a direct route for human exposure [21].

The toxicity of heavy metals is exacerbated by their strong affinity for cellular components such as proteins and DNA, leading to enzymatic dysfunction and the induction of oxidative stress. Their presence in the environment thus represents a critical challenge for maintaining ecological stability and protecting public health. Accurate monitoring of heavy metals is crucial for identifying contamination sources, evaluating exposure risks, and ensuring compliance with environmental regulations. Advanced detection methods, including electrochemical sensing with BDDEs, provide highly sensitive, reliable, and rapid approaches for detecting heavy metals in water, soil, and food matrices (Table 1). Such monitoring efforts are vital not only for addressing immediate health concerns but also for preventing long-term environmental damage and preserving ecosystems for future generations [22].

### 3.1. Cd^2+^

Cadmium (Cd) is a highly hazardous heavy metal that presents substantial risks to both human health and the environment. It predominantly accumulates in the kidneys, impairing their filtration capacity and contributing to chronic conditions such as osteoporosis and renal failure. Recognized as a Group 1 carcinogen by the International Agency for Research on Cancer (IARC), cadmium exposure is strongly associated with an increased risk of lung and kidney cancers, as well as reproductive toxicity. Furthermore, its affinity to induce oxidative stress disrupts critical cellular processes, resulting in inflammation and DNA damage [23].

The use of BDDE for Cd monitoring in aqueous matrices is illustrated in the article published by McGaw et al. Anodic stripping voltammetry was the method used, in which Cd ions were electrochemically stripped for quantification after first being preconcentrated on the electrode surface. The high sensitivity of BDDEs was demonstrated by the detection limit that was attained, which was in the parts-per-billion (ppb) level [24].

One noteworthy difference is that BDDEs overcome the drawbacks of mercury-based systems by demonstrating resistance to fouling and offering repeatable findings over several measurements and electrodes. BDDEs eliminate problems with mercury’s toxicity and volatility and do not require as much surface pretreatment as Hg electrodes do. The promise of BDDEs for trustworthy environmental monitoring and heavy metal detection is highlighted by their flexibility in harsh conditions and compatibility with actual sample matrices [25].

Adam Moris et al. developed an interesting method for detecting Cd^2+^ that uses a vibrating BDDE to perform better than traditional methods. When vibration was applied, mass transfer to the electrode surface was greatly enhanced, and at a frequency of 133 Hz, the reduction reaction current density increased by 92.6% in comparison to a static arrangement. For Cd^2+^ concentrations of 10 mg/L, this novel method increased the peak current during square-wave anodic stripping voltammetry (SWASV) by a factor of 5.3, allowing for extremely sensitive detection with the LOD and LOQ of 0.04 μg L^−1^ and 0.12 μg L^−1^, respectively [26].

### 3.2. Cu^2+^

While copper ions (Cu^2+^) are essential for various biological processes, they become toxic at rising concentrations. Copper toxicity arises from the generation of reactive oxygen species (ROS), which induce oxidative stress, leading to damage to proteins, lipids, and DNA. Excess Cu^2+^ adversely affects aquatic ecosystems by impairing fish gill function and reducing biodiversity, while in humans, it is associated with neurological damage, gastrointestinal distress, and hepatic toxicity. Moreover, the bioaccumulation of copper in living organisms poses a significant risk to the food chain. Effective monitoring of Cu^2+^ concentrations is crucial to mitigating their detrimental effects on both environmental and human health [27,28].

Sonthalia et al. emphasized how copper ions can be found in a variety of environmental samples, such as tap water, lake water, and well water, using boron-doped nanocrystalline diamond thin-film electrodes. These electrodes’ exceptional sensitivity and repeatability, which enable a linear detection range for Cu^2+^ concentrations ranging from 1 to 130 ppb, is a significant advance. The accuracy and dependability of the electrodes were confirmed by the results, which showed outstanding agreement with conventional techniques, including atomic absorption spectrometry and inductively coupled plasma mass spectrometry (ICP-MS) [29].

Interestingly, researchers demonstrated the ability to simultaneously detect Cu^2+^ and Pb^2+^ using BDD electrodes, enabling independent analysis of both metals without mutual interference. This approach relies on distinct oxidation peaks for each metal, allowing precise quantification even in complex samples [30].

### 3.3. As^3+^

Arsenic (As^3+^) ions are highly poisonous and represent serious dangers to both human health and the environment. They impair enzyme activity and cause oxidative stress, which destroys DNA, proteins, and lipids. Long-term exposure to As^3+^ can lead to chronic illnesses such as skin lesions, cancer, and organ damage, while acute exposure can cause severe gastrointestinal symptoms and multi-organ failure. As^3+^ is released into the environment by natural processes and industrial operations, damaging water, soil, and the food chain. Because of its high bioavailability and toxicity, monitoring and mitigating arsenic contamination is a top priority for human health and the environment [31,32].

The investigation conducted by Hrapovic and colleagues illustrates how a platinum nanoparticle-modified BDDE may detect As^3+^ through oxidation. The approach has a detection limit as low as 0.5 ppb and is highly sensitive and stable even in the presence of major interferents such as copper and chloride ions, which are frequent in groundwater. Another remarkable characteristic is the electrode’s reusability, which is achieved through electrochemical etching of the platinum layer, resulting in consistent performance across 150 repeating runs. This innovation demonstrates the potential of BDDEs for precise and dependable arsenic monitoring in complicated environmental samples [33].

### 3.4. Pb^2+^

The environment and human health are seriously endangered by lead ions (Pb^2+^), which are extremely poisonous. By attaching themselves to sulfhydryl groups in proteins and taking the place of vital metals like calcium, zinc, and iron in biological systems, they disrupt enzymatic functions. Mining, industrial operations, and the usage of lead-based products all contribute to the environmental contamination of water, soil, and air by Pb^2+^. Its tenacity and bioaccumulation in ecosystems highlight how urgently monitoring and mitigation are needed to safeguard biodiversity and public health [34,35].

It is noteworthy to highlight that in the cited research, co-deposited lead and silver ion interactions were examined using a BDDE during ASV. The study found that peak broadening and splitting are caused by bonding interactions between Pb and Ag on the electrode surface, which has a considerable impact on stripping voltammetry results when these metals are deposited simultaneously. Another noteworthy discovery is that when Ag deposits cover Pb, encapsulation effects take place, which lowers the stripping yield of lead at its typical potential. These results demonstrate the complexity of multi-metal systems and the requirement for sophisticated calibrating techniques to guarantee precise detection in practical applications [36].

Another study examined the influence of temperature on the electrodeposition and anodic stripping of Pb utilizing BDDEs. The findings revealed that increasing the temperature from 20 °C to 60 °C significantly enhanced the deposition efficiency, resulting in a tenfold increase in the stripping signal and a notable decrease in the detection limit. Additionally, increased temperatures were observed to accelerate the nucleation and growth of lead deposits on the electrode surface, as confirmed by atomic force microscopy (AFM), which demonstrated the formation of larger and more coalesced lead clusters. These results underscore the importance of temperature optimization in enhancing the sensitivity and overall performance of Pb^2+^ detection using BDDEs [37].

### 3.5. Hg^2+^

Particularly in its methylmercury form, mercury (Hg) is extremely poisonous and dangerous for both human health and the environment. Through bioaccumulation and biomagnification, it builds up in living things, increasing concentrations in top predators like humans. Mercury exposure primarily affects the nervous system, causing cognitive impairments, motor dysfunction, and developmental delays, especially in children. Chronic exposure can also damage the kidneys and cardiovascular system, while acute poisoning may result in severe respiratory failure. The persistence and mobility of mercury in the environment make it a critical pollutant, requiring strict monitoring and mitigation efforts [38,39].

However, in the study conducted by Manivannan, A. et al., a unique technique for Hg^2+^ detection at the parts-per-billion level utilizing BDDEs was proposed. The researchers discovered that co-depositing minor amounts of gold (1–3 ppm) on the BDDE surface improved the method’s sensitivity and repeatability by inhibiting the production of insoluble mercurous chloride (calomel), which frequently interferes with results. Another important finding is that this method displayed excellent linearity in the detection range of 2–10 ppb and was then applied to environmental samples, demonstrating its potential for dependable on-site mercury monitoring [40].

It is worthwhile to mention that in the cited investigation, the use of BDDE in conjunction with a rotating disk electrode (RDE) approach improved the detection of Hg^2+^. Even in complicated matrices like flue gas samples, the method allowed for sensitive detection down to the parts-per-trillion level by greatly enhancing the electrolyte’s hydrodynamic movement.

In order to ensure precision and reproducibility in chloride-based media, tiny amounts of gold (3 ppm) are added to the electrode surface to prevent the production of insoluble calomel. These developments demonstrate the method’s potential for industrial applications and on-site environmental monitoring [41].

**Table 1 sensors-25-02339-t001:** Summary of analytical information for metal ion identification and determination techniques.

Heavy Metal	Linear Range (µM)	LOD (µM)	Technique	Ref.
Cd^2+^	0–0.89	0.0347	ASV	[24]
	0–4.45	0.0267	ASV	[25]
	0.00009–0.0089	0.00036	SWASV	[26]
Cu^2+^	0.01574–2.0453	/	ASV	[29]
	2.5–100	0.5	SWV	[30]
As^3+^	0–4.00	0.534	LSV *	[33]
Pb^2+^	25–750	/	ASV	[36]
Hg^2+^	0.01–0.05	0.002	DPV	[40]
	0.00025–0.25	0.015	DPASV *	[41]

Abbreviations: DPASV—differential pulse anodic stripping voltammetry; LSV—linear sweep voltammetry; ASV—Anodic Stripping Voltammetry; SWASV—Square Wave Anodic Stripping Voltammetry; DPV—Differential pulse voltammetry.

## 4. Industrial Pollutants: Hidden Hazards and Modern Solutions

Industrial pollutants, particularly organic substances, represent a significant threat to both human health and the environment. These pollutants include a wide range of chemicals, such as phenols, polycyclic aromatic hydrocarbons (PAHs), dyes, and volatile organic compounds (VOCs), which are released during industrial processes like chemical manufacturing, textile production, and oil refining. They are commonly found in industrial wastewater, contaminated soil, and air emissions, posing risks to ecosystems and nearby communities. The impact of organic pollutants on human health is profound. Many of these compounds are toxic, carcinogenic, or mutagenic, and prolonged exposure can lead to severe health issues, including respiratory problems, liver and kidney damage, and hormonal disruptions. Organic pollutants damage fish and other aquatic life by lowering oxygen levels in the water, upsetting aquatic ecosystems. Plant growth and agricultural productivity are also impacted by the degradation of soil quality. Airborne organic pollutants, such as volatile organic compounds, cause air pollution and climate change by forming ground-level ozone and smog [42,43].

The importance of monitoring organic industrial pollutants cannot be overstated. Early detection and accurate quantification are essential to prevent their spread and mitigate their effects. BDDEs enable sensitive detection of organic substances, even in complex environmental matrices like industrial effluents [44].

### 4.1. Chlorophenols

Chlorophenols (CPs) are highly toxic organic compounds commonly used in pesticides, disinfectants, and industrial processes such as wood preservation and dye manufacturing. They can contaminate water, soil, and air, posing significant risks to ecosystems and human health. In the environment, chlorophenols disrupt aquatic ecosystems, reduce biodiversity, and bioaccumulate in organisms, affecting food chains. Monitoring and preventing chlorophenol contamination are critical to mitigate their harmful effects, protect public health, and maintain ecological balance [45,46].

Terashima, C. and colleagues used an anodically pretreated BDDE for the sensitive detection of chlorophenols in environmental water samples. The electrode exhibited exceptional resistance to fouling, even at high CP concentrations, due to the presence of anodically generated oxygen functional groups that repel phenoxy radicals. Another notable feature is the integration of a BDDE with high-performance liquid chromatography (HPLC) and flow injection analysis (FIA), achieving detection limits as low as 0.4 nM through an innovative column-switching technique that enhances preconcentration [47].

The electrochemical oxidation of 4-chlorophenol (4-CP) was investigated using a BDDE thin-film electrode in acidic media. The results demonstrated that at high anodic potentials, hydroxyl radicals generated on the BDDE surface enabled the complete mineralization of 4-CP into carbon dioxide, effectively preventing electrode fouling caused by polymeric byproducts.

Another key finding is the development of a theoretical model predicting the evolution of chemical oxygen demand (COD) and current efficiency during the oxidation process, which aligns well with experimental data.

These insights highlight the robustness of BDDEs for treating refractory organic pollutants in industrial wastewater [48].

### 4.2. Triazole Fungicides

Although triazole fungicides are frequently employed in agriculture to manage fungal diseases in crops, they are organic pollutants in the environment due to their toxicity and persistence. They are frequently found in soils, surface waters, and agricultural runoff, where they can build up and affect organisms that are not the intended targets. The toxicity of triazole fungicides is linked to their ability to inhibit cytochrome P450 enzymes, disrupting key metabolic pathways in plants, animals, and microorganisms [49,50].

Monitoring itraconazole is crucial due to its widespread use as an antifungal agent, its persistence in the environment, and its potential to disrupt aquatic ecosystems and non-target organisms through bioaccumulation and toxicity. Mielech-Łukasiewicz et al. detected itraconazole using a BDDE and SWV, enabling high sensitivity and precision in the analysis of pharmaceutical contaminants. The experiments were optimized for pH and voltammetry parameters, achieving a detection limit of 1.79 × 10^−8^ mol/L for itraconazole. The method was applied to real water samples, including river and tap water, with high accuracy and recovery rates between 93.5% and 101%, demonstrating the practical application of BDDE for monitoring pharmaceutical pollutants in the environment [51].

The cited research shows that using a BDDE in conjunction with DPV to detect difenoconazole and paclobutrazol simultaneously. Significantly, the BDDE demonstrated outstanding resilience to surface fouling and enabled detection at high oxidation potentials (about +1675 mV for difenoconazole), guaranteeing precise analysis in intricate matrices such as pesticide formulations and tainted river water [52].

### 4.3. Hydroquinone

Due to its extensive use as a reducing agent, skin-lightening agent, and antioxidant in sectors like chemical manufacturing, photography, and cosmetics, hydroquinone is an important environmental pollutant. Because of its toxicity and capacity to produce reactive oxygen species, it is frequently found in contaminated water sources and industrial effluents, harming aquatic ecosystems as well as human health. Because of their high sensitivity, resistance to fouling, and capacity to function in complex matrices such as industrial effluent, BDDEs are perfect for the detection of hydroquinone, which must be monitored to avoid its harmful effects [53,54].

The redox kinetics of hydroquinone (HQ) and quinone (Q) were examined in the study conducted by Ramesham, R. et al., utilizing cyclic voltammetry and a BDDE, with an emphasis on their electrochemical irreversibility. Notably, the BDDE demonstrated outstanding stability throughout a wide potential range (−2.0 V to +1.4 V vs. SHE), allowing for precise and reproducible detection of HQ/Q even in complex acidic and neutral conditions. These qualities highlight BDDEs’ ability to identify organic contaminants such as hydroquinone in demanding environmental matrices [55].

### 4.4. Hydrazine

Because of its highly toxic nature and widespread use as a corrosion inhibitor in the pharmaceutical, water treatment, and rocket propulsion industries, hydrazine is a notable environmental contaminant. It is commonly found in industrial effluents and presents serious hazards to human health and aquatic ecosystems, including the possibility of cancer and harm to critical organs [56].

In the study conducted by Sun et al., a BDDE was employed for the detection of hydrazine via direct electrochemical oxidation utilizing DPV. The BDDE demonstrated an extensive potential window and minimal background current, facilitating the sensitive quantification of hydrazine with a detection limit of 1 µM in neutral pH environments. These characteristics highlight the effectiveness of BDDEs for dependable hydrazine monitoring in both environmental and industrial contexts [57].

### 4.5. Cresols

Cresols, widely utilized in industries such as petrochemicals, pharmaceuticals, and wood preservation, are recognized as hazardous environmental pollutants due to their high toxicity. Commonly detected in industrial effluents and contaminated soils, they present significant risks to both aquatic ecosystems and human health, contributing to liver damage, neurological disorders, and respiratory issues [58,59,60].

Flox et al. investigated the electrochemical oxidation and subsequent degradation of cresols using BDDEs, concentrating on their mineralization into CO_2_ and H_2_O. While cyclic voltammetry was used to clarify the underlying degradation mechanisms under various operating parameters, the investigation focused on the generation of quasi-free hydroxyl radicals on the BDDE surface, which greatly accelerated the oxidation process [61].

On the other hand, a BDDE was applied for the sensitive detection and quantification of cresols in contaminated water samples using differential pulse voltammetry. This method achieved high sensitivity, reproducibility, and resistance to fouling, ensuring precise monitoring in complex wastewater matrices. Together, these studies highlight the versatility of BDDE for both the monitoring and treatment of cresol pollution, advancing its application in environmental remediation [62].

For the sensitive detection, dependable monitoring, and effective degradation of cresols in environmental applications, the combined results of the two investigations highlight the efficiency and adaptability of BDDEs.

### 4.6. Pigments and Dyes

Because of their extensive use in sectors including textiles, printing, and plastics, as well as their durability in both aquatic and terrestrial ecosystems, pigments and dyes constitute major environmental contaminants. Through processes including bioaccumulation and the poisoning of potable water supplies, these compounds pose serious threats to aquatic ecosystems and human health since they are often poisonous, carcinogenic, and resistant to biodegradation [63,64].

Methyl orange (MO) is recognized as a hazardous pollutant due to its extensive application in the textile and printing industries, as well as its persistence and toxicity in aquatic ecosystems. In the cited study, Chen et al. explored the degradation of MO using BDDEs, with particular emphasis on their capacity to achieve complete mineralization of MO into CO_2_ and water. The BDDE demonstrated a high oxygen evolution potential (approximately 2.0 V vs. SCE), enabling the efficient oxidation of MO through the generation of hydroxyl radicals. Moreover, the BDDE exhibited remarkable stability under a range of pH conditions and with various supporting electrolytes, facilitating the effective degradation of MO without requiring prior pH adjustment [65].

## 5. Advanced Pesticide Detection with BDDE Sensors: Precision and Reliability

Although pesticides are necessary in modern agriculture to control pests and boost crop yields, their widespread use poses serious risks to both human health and the environment. Numerous pesticides persist in the environment, bioaccumulate in living creatures, and contaminate soil, water, and air, endangering non-target species like pollinators, aquatic life, and even people. Long-term exposure to some pesticides has been linked to neurotoxicity, carcinogenicity, and endocrine disruption, highlighting the importance of keeping an eye on their environmental presence [66].

Advanced detection methods are crucial for properly tracking pesticide contamination, maintaining regulatory compliance, and conserving ecosystems. BDDEs are ideal for pesticide detection because of their excellent sensitivity, chemical stability, and resistance to fouling (Table 2). Their use in the development of robust, dependable electrochemical sensors is critical for improving pesticide monitoring and reducing their environmental impact [10].

### 5.1. BDDE Electrodes: Enhancing Carbamate Pesticide Detection

Carbamate pesticides are widely used in agriculture to manage a variety of pests, including insects, weeds, and fungi, due to their high efficacy and short environmental persistence when compared to other pesticide categories. They work by inhibiting acetylcholinesterase, an enzyme essential for nervous system function, making them exceedingly toxic to target pests but potentially harmful to non-target animals, including humans. Overexposure can result in immediate and long-term health consequences, including brain damage. Carbamates frequently contaminate water sources via agricultural runoff, posing significant risks to aquatic ecosystems and wildlife.

Regular monitoring of their concentrations in the environment is crucial for preventing bioaccumulation and ensuring the safety of water, soil, and food supplies [67,68].

A BDDE was employed to detect and measure formetanate (FMT), a carbamate insecticide, in mango and grape samples using square-wave voltammetry. Ribeiro et al. optimized the method at a pH of 7.0, achieving a detection limit of 3.7 × 10^−7^ mol/L. Sensitivity and repeatability were enhanced through cathodic pretreatment of the electrode. Importantly, the unmodified BDDE demonstrated high resistance to fouling, making the process simpler and more cost-effective compared to modified sensors. The technique showed excellent accuracy, with recovery rates between 95.2% and 105.2% in complex fruit samples, confirming its reliability and efficiency for pesticide monitoring in food safety [69].

A BDDE was applied for the electrochemical detection of the carbamate insecticide methomyl using SWV and DPV. The analysis conducted by Costa et al. showed that BDDE was conducted on real-world samples, including tap water, river water, and commercial pesticide formulations. A notable aspect of this study was the use of the cathodically pretreated BDDE, which enhanced sensitivity, minimized background currents, and mitigated interference from complex sample matrices. By achieving a low detection limit of 1.2 × 10^−6^ mol/L through DPV, the study demonstrated the electrode’s capability for precise and efficient monitoring of pesticides in both environmental and commercial samples [70].

In research conducted by Codognoto et al., SWV combined with a BDDE enabled the direct detection of carbaryl in natural water samples. The method achieved a detection limit of 8.2 µg/L at elevated oxidation potentials (+1.42 V vs. Ag/AgCl) under slightly acidic conditions (pH 6.0), eliminating the need for pretreatment steps such as extraction or preconcentration. Notably, the technique demonstrated robust performance even in contaminated water samples, with minimal interference from organic matter or coexisting pesticides, including 4-nitrophenol and fenthion.

In a parallel study conducted by Pop et al., a graphene-modified BDDE was employed to simultaneously detect carbaryl and paraquat using CV and DPV in both buffer solutions and real-world matrices, such as fresh apple juice. The graphene modification significantly enhanced the electrooxidation signal of carbaryl, lowering the detection limit to 70 nM at a reduced potential of +0.74 V vs. SCE. This approach demonstrated high sensitivity and reliability in complex matrices without requiring pretreatment, underscoring the versatility and efficacy of BDDE for advanced pesticide monitoring applications [71,72].

A BDDE was utilized for the detection and quantification of maneb in the research conducted by Stankovic et al., employing DPV and CV to explore its electrochemical behavior. The analysis was performed on real samples, including spiked river water, achieving a low detection limit of 7.2 nM. The study demonstrated the robustness of the BDDE, emphasizing its low background current and high resistance to fouling, which are critical for accurate detection in complex environmental matrices. Anodic pretreatment of the electrode further enhanced its performance by increasing the number of oxygen-containing surface functional groups, which improved electron transfer kinetics and reduced background currents. This modification enabled more sensitive and reproducible measurements, particularly in complex environmental samples. This work highlights the advantages of BDDEs as a reliable, stable, and environmentally friendly alternative to conventional electrodes, particularly for long-term monitoring of pesticide residues in aquatic environments [73].

Different pretreatment methods of BDDEs (cathodic and anodic) enable a diverse range of applications in pollutant detection. The scope of use for BDDEs with various pretreatments is presented in Figure 3.

### 5.2. High-Precision Detection of Organophosphates Using BDDEs

Organophosphates are commonly used in agriculture as highly effective insecticides that control a wide range of pests while also protecting crops and enhancing crop harvests. Their mechanism of operation involves inhibiting acetylcholinesterase, a critical enzyme in the nervous system, making them extremely dangerous to both pests and non-target animals like humans and wildlife. These compounds are known to contaminate water and soil via wastewater from farms, posing a hazard to aquatic habitats and wildlife. Prolonged exposure to organophosphates in humans can result in neurological disorders, respiratory problems, and developmental abnormalities [67].

Without the need for preconcentration or derivation procedures, França et al. used a BDDE in conjunction with SWV to detect fenthion in Passiflora alata tinctures. An irreversible oxidation peak that was consistent across pH levels was found at +1.27 V vs. Ag/AgCl. The technique demonstrated outstanding resistance to fouling in a complex herbal matrix and exhibited great sensitivity, with a detection limit of 21 µg/L. This method provides a quick, dependable, and eco-friendly substitute for tracking pesticides in items made from therapeutic plants [74].

It is interesting how Vukojević Vesna et al. used a BDDE with SWV to detect the organophosphorus insecticide azamethiphos. In 1 M nitric acid, a high oxidation peak at +1.7 V vs. Ag/AgCl verified the compound’s electrochemical irreversibility. With little sample preparation, the technique showed a broad linear range of 2–100 µM and a detection limit of 0.45 µM. For monitoring pesticides in environmental water samples, the electrode is a dependable and affordable substitute for conventional chromatographic techniques due to its high selectivity and fouling resistance [75].

### 5.3. Electrochemical Detection of Diverse Pesticide Classes Using BDDEs

Bentazone was found in river water samples from Timok and Ibar in research conducted by Jevtić Sonja et al. using a BDDE and DPV. An oxidation peak was seen at +1.0 V in Britton–Robinson buffer (pH 4). The technique showed linearity in the 2–100 µM range and reached a detection limit of 0.5 µM. The use of unmodified BDDE produced repeatable results in dirty water samples, maintained good fouling resistance, and avoided the requirement for surface modifications. This straightforward, reasonably priced, and extremely sensitive method offers a viable way to monitor bentazone in environmental settings [76].

In order to identify diuron in water samples using DPV, Duarte et al. developed the electroanalytical method by using a BDDE as a working electrode. This method showed outstanding resistance to interference from other herbicides and organic materials, and it was able to attain a detection limit of 0.035 µmol/L. The innovation is in combining preconcentration with a BDDE, which outperformed traditional techniques in terms of sensitivity and enabled precise diuron detection in intricate environmental matrices [77].

Another study conducted by Lima et al. employed an electrochemical detection approach for chlorothalonil, an organochlorine fungicide, with a BDDE and SWV. In the investigations, chlorothalonil demonstrated three distinct reduction peaks at −1.07, −1.2, and −1.4 V (vs. Ag/AgCl) in Britton–Robinson buffer (pH 8.0) using genuine tea infusion samples. Density functional theory (DFT) studies confirmed the reduction mechanism’s sequential dehalogenation steps, shedding light on chemical alterations. With a detection limit of 10.6 µg/L, the technique demonstrated good sensitivity, making it a reliable tool for monitoring food and environmental safety [78].

Barek et al. used DPV to detect 2-aminobiphenyl, a genotoxic pollutant from cigarette smoke and combustion byproducts, using boron-doped diamond nanocrystalline film electrodes (BDDNDFE). Britton–Robinson buffer at pH 7 was used to optimize detection, and an oxidation peak at +0.85 V vs. Ag/AgCl was observed. Even in complicated sample matrices, the BDDNDFE showed exceptional stability and fouling resistance, with a detection limit of 1.2 × 10^−7^ mol/L. This demonstrates how well the electrode monitors dangerous contaminants in biological and environmental settings compared to conventional carbon-based systems [79].

**Table 2 sensors-25-02339-t002:** A concise overview of the analytical parameters used to detect several pesticide classes on BDDEs.

Pesticide	Class of Pesticide	LOD (µM)	Technique	Type of Electrode Modifier	Linear Range (µM)	Ref.
Formetanate	Carbamate	0.37	SWV	CPT *	4.98–17	[69]
Methomyl		1.2	DPV	CPT *	5–41	[70]
Carbaryl		40.8	SWV	/	2.5–30	[71]
		0.07	DPV	Graphene-modified	0.2–12	[72]
Maneb		0.072	DPV	APT *	0.08–3.0	[73]
Fenthion	Organophosphates	0.0755	SWV	/	1–10	[74]
Azamethiphos		0.45	SWV	/	2–100	[75]
Bentazone	Thiadiazine herbicide	0.5	DPV	/	2–100	[76]
Diuron	Phenylurea herbicide	0.035	DPV	/	1–9	[77]
Chlorothalonil	Organochlorine fungicide	0.04	SWV	/	0.12–4	[78]
2-aminobiphenyl	Biphenyl derivative	0.12	DPV	Nanocrystalline film	0.1–10	[79]

Abbreviations: CPT—cathodically pretreated; APT—anodically pretreated; SWV—Square wave voltammetry; DPV—Differential pulse voltammetry.

## 6. Electrochemical Detection of Pharmaceuticals with BDDEs

Pharmaceuticals have emerged as significant environmental contaminants due to their widespread use and poor disposal, resulting in their discovery in a variety of environmental compartments such as water, soil, and air. Following human intake, a fraction of these compounds are voided in unmetabolized form and enter wastewater treatment systems, which frequently fail to remove them completely. Consequently, pharmaceutical residues are frequently discovered in surface water, groundwater, and even drinking water. Their existence threatens aquatic ecosystems by affecting aquatic animals’ behavior, reproduction, and survival. Antibiotics are particularly concerning since they contribute to the spread of antibiotic-resistant microorganisms, posing a serious public health threat [80].

The persistence of certain pharmaceuticals in the environment allows them to bioaccumulate in organisms, potentially impacting food chains. Hormonal drugs, such as contraceptives, are known to cause endocrine disruption in aquatic species, leading to altered reproductive patterns. Anti-inflammatory drugs, like diclofenac, have been linked to organ damage in fish and birds. Moreover, the presence of pharmaceuticals in agricultural soils, resulting from the use of treated wastewater or sludge as fertilizer, can affect soil microbial communities and crop growth. Effective monitoring and proper management of pharmaceutical waste are crucial to mitigating their long-term environmental impact [81].

### 6.1. Detection of Analgesics and Antipyretics Using BDDEs

Analgesics and antipyretics, such as ibuprofen, paracetamol, and diclofenac, are among the most widely used pharmaceuticals, with annual global consumption reaching tens of thousands of tons. Their extensive use in medicine and over-the-counter treatments leads to frequent presence in wastewater, as they are partially excreted unmetabolized through urine and feces. In the environment, these drugs can cause toxic effects on aquatic organisms, including liver and kidney damage in fish, and reduce biodiversity in aquatic ecosystems [82].

Nattakarn et al. developed an electrochemical method for the detection of acetaminophen using a BDDE combined with cyclic voltammetry (CV) and flow injection amperometric detection. This approach achieved a linear detection range of 0.5 to 50 µM with a detection limit of 10 nM. The method was successfully applied to pharmaceutical syrup samples, demonstrating high sensitivity and precision in real-world matrices [83].

In a different work, Wang et al. described a unique hybrid sensor made of nanometer-sized graphite and a BDDE and produced via chemical vapor deposition. This sensor, using differential normal pulse voltammetry (DNPV), has an expanded linear range of 0.02 to 50 µM and a detection limit of 5 nM. The hybrid electrode’s increased surface area and improved electron transfer qualities resulted in superior selectivity, repeatability, and fouling resistance. The approach was used on water and biological samples, demonstrating its potential for enhanced pharmaceutical monitoring [84].

Using a cathodically pretreated BDDE, Konan Martin and colleagues created an electroanalytical technique for the detection of paracetamol in pharmaceutical formulations. The oxidation of paracetamol in HClO_4_ electrolyte was measured using differential pulse voltammetry (DPV) and cyclic voltammetry (CV), with a reported detection limit of 0.16 µM. The study highlighted BDDEs’ exceptional fouling resistance, which guarantees great sensitivity and repeatability in intricate pharmaceutical samples. This method showed how BDDEs may be used for trustworthy pharmaceutical quality monitoring and control [85].

In contrast, Niedziałkowski et al. explored paracetamol detection by comparing BDDEs with boron-doped carbon nanowall electrodes (B:CNW). The BDDE demonstrated an oxidation peak at +0.47 V in a phosphate buffer solution (pH 7.0) and achieved a detection limit of 0.43 µM. Notably, the unmodified BDDE exhibited excellent stability and reproducibility, proving to be a reliable option for paracetamol detection in both environmental and biological samples. This study underscored the adaptability of BDDEs for analyzing complex sample matrices without necessitating surface modifications, highlighting its potential for broad analytical applications [86].

In their research, Kouadio et al. applied BDDEs for the detection of paracetamol in real environmental samples, focusing on river water analysis. DPV was used as the primary technique, conducted in phosphate buffer at neutral pH, achieving an impressive detection limit of 0.065 µM. The study highlighted the strong linearity and high sensitivity of the method, with BDDEs showing excellent performance in handling complex environmental matrices. A key innovation of this work is its successful application in real-world water monitoring, showcasing the potential of BDDEs for reliable environmental pollutant detection [87].

Lima et al. investigated the electrochemical oxidation and determination of ibuprofen using BDDEs. They employed differential pulse voltammetry (DPV) in 0.1 M sulfuric acid with 10% ethanol, observing an irreversible oxidation peak around +1.65 V. The detection limit achieved was 5 µM, and the method showed a linear range from 20 to 400 µM. The study’s novelty lies in the electrode’s anodic pretreatment, which enhanced the sensitivity and reproducibility of ibuprofen detection. The approach was successfully applied to pharmaceutical formulations, highlighting BDDEs’ suitability for sensitive drug analysis in complex matrices [88].

Švorc and colleagues presented an advanced electroanalytical approach using a bare, electrochemically untreated BDDE for ibuprofen detection in pharmaceutical products and spiked human urine samples. CV and DPV were utilized, with an oxidation peak observed at +1.75 V in 1 M perchloric acid. The method demonstrated detection limits of 0.41 µM (DPV) and 0.93 µM (square-wave voltammetry, SWV). Compared to previous works, the novelty here is the use of an untreated BDDE, eliminating the need for pretreatment while ensuring high precision and minimal matrix interference. The study confirmed the protocol’s reliability for routine pharmaceutical and biological sample analysis [89].

Lucas et al. evaluated the electrochemical detection of diclofenac using a BDDE combined with square-wave voltammetry. The approach showed great sensitivity, with a detection limit of 1.15 × 10^−7^ mol/L and a linear response range of 4.94 × 10^−7^ to 4.43 × 10^−6^ mol/L. Real samples were analyzed, such as prescription tablets and synthetic urine. The electrode’s resistance to fouling, great repeatability, and little sample preparation were important aspects of the study since they provided a more efficient and cost-effective alternative to traditional chromatographic procedures. Furthermore, computational modeling and UV–Vis spectroscopy were used to analyze diclofenac’s oxidation mechanism, which revealed dimer formation and provided new insights into its electrochemical behavior [90].

### 6.2. Electrochemical Monitoring of Antibiotics Using BDDEs

Antibacterial drugs, particularly antibiotics, are significant environmental pollutants due to their widespread use in human and veterinary medicine, with annual global consumption exceeding tens of thousands of tons. A large portion of antibiotics is excreted unmetabolized, entering wastewater systems and ultimately contaminating surface water, soil, and even groundwater. Their presence in the environment can disrupt microbial communities, promote the development of antibiotic-resistant bacteria, and harm aquatic ecosystems. Antibiotics in low concentrations have been shown to exert selective pressure on bacteria, accelerating the spread of resistance genes, which poses a serious threat to public health. Monitoring and proper management of antibiotic residues are crucial to mitigating their environmental and health impacts [91].

Radičová et al. used BDDEs to detect ciprofloxacin at different doses. They used SWV in an ammonium acetate buffer at pH 5 to discover a well-defined irreversible oxidation peak at +1.15 V vs. Ag/AgCl. The approach obtained a low detection limit of 0.05 µM and was applied to real urine samples, with recovery rates ranging from 97% to 102%. The study stresses the influence of boron content on electrode sensitivity and BDDEs’ potential for pharmacological monitoring in biological matrices [92].

The impact of boron content on the efficacy of BDDEs for ciprofloxacin detection was investigated by Ilmi Nur Indriani Savitri et al. In phosphate buffer (pH 7), the study used SWV and obtained a detection limit of 0.17 µM. Wastewater, milk, and pharmaceutical tablet samples were all successfully treated with the technique. The methodical tuning of boron content was a crucial component of this study, as it significantly improved the overall analytical performance by influencing both sensitivity and detection limits [93].

In order to remove ciprofloxacin from a variety of aqueous matrices, such as synthetic urine and actual wastewater, Alexsandro J. dos Santos et al. compared microcrystalline and nanocrystalline BDDEs. According to their research, nanocrystalline BDDEs completely degraded in solutions containing chloride, demonstrating better electrocatalytic efficacy than microcrystalline electrodes. This work illustrates how BDDE shape plays a crucial role in improving oxidation efficiency and shows how it may be used to handle complicated real-world materials [94].

Ennouri et al. investigated the electrochemical degradation of chloramphenicol, a broad-spectrum antibiotic, using a BDDE combined with UV irradiation. The study employed an advanced oxidation process (AOP) where UV-assisted anodic oxidation facilitated the generation of hydroxyl radicals and other reactive species, enhancing the degradation and mineralization of chloramphenicol in water. Notably, a thin-film BDDE was used, and optimal performance was achieved at pH 10 with a current density of 300 mA m^−2^, leading to the complete removal of the antibiotic within 150 min and 95% mineralization after 3 h [95].

The use of BDDEs for the detection of chloramphenicol and ciprofloxacin has proven to be highly effective due to their exceptional electrochemical properties, including a wide potential window, low background current, and high resistance to fouling. BDDEs enable sensitive and reliable quantification of these antibiotics in complex matrices, such as pharmaceutical formulations, biological fluids, and environmental water samples.

### 6.3. Sensitive Electrochemical Detection of Endocrine Disruptors Using BDDEs

Because of their extensive usage and durability, chemicals having endocrine-disrupting effects—also referred to as endocrine disruptors—are important environmental pollutants. Millions of tons of these chemicals are produced worldwide each year and are present in a wide range of items, including cosmetics, medications, and plastics (BPA, phthalates). Both humans and wildlife suffer from developmental, reproductive, neurological, and immune system diseases as a result of endocrine disruptors interfering with their hormonal systems. Because of their propensity to bioaccumulate and persist in the environment, these compounds must be monitored and regulated in order to protect biodiversity and public health [96].

By employing a BDDE in galvanostatic mode, Murugananthan et al. investigated the anodic oxidation of bisphenol A (BPA). The electrochemical behavior of BPA was examined using cyclic voltammetry, which showed an oxidation peak in a Na_2_SO_4_ electrolyte at +1.2 V vs. SCE. Due to the production of hydroxyl radicals on the electrode surface, the study showed that the BDDE efficiently promoted the full mineralization of BPA. A sustainable and efficient technique for BPA removal in aqueous environments was presented in this study, which notably included a thorough examination of the kinetics and current efficiency involved in BPA degradation [97].

Using an unmodified BDDE and DPV, Pereira et al. created a technique for the detection of bisphenol A (BPA). Cathodic pretreatment of the electrode increased sensitivity, resulting in a detection limit of 0.21 µmol/L. The use of the unmodified BDDE for BPA detection across many matrices, exhibiting strong robustness and little interference in complicated materials, is what makes this study interesting. Furthermore, the approach is a workable solution for regular environmental monitoring due to its simplicity and dependability [98].

In their study, Wang and Farrell investigated the electrochemical inactivation of triclosan in aqueous solutions using BDDE film electrodes. LSV and CV were employed, with triclosan oxidation initiating at approximately +0.4 V vs. SHE. The findings indicated that direct oxidation of triclosan occurred at the electrode surface below 2 V, while indirect oxidation became predominant at higher potentials. Notably, the study demonstrated that operating at potentials above 3 V prevented electrode passivation, enabling the complete oxidation of triclosan to CO_2_. The key innovation of this research lies in achieving efficient triclosan degradation at high current densities without generating persistent toxic byproducts, offering a promising strategy for advanced water treatment technologies [99].

Using an unaltered BDDE, Schmitz et al. created a sensitive and dependable electroanalytical technique to identify triclocarban, a substance chemically similar to triclosan. Triclocarban was detected using DPV, which produced oxidation peaks at +1.10 V and +1.40 V vs. Ag/AgCl. The technique was used on actual samples of antibacterial soaps and river water, and the study obtained a detection limit of 18 nM. The novel aspect of this work is the cathodic pretreatment of the BDDE, which improves sensitivity and preserves electrode performance throughout several measurements and the low amount of sample preparation needed [100].

### 6.4. Detection of Various Pharmaceutical Classes Using BDDEs

Detecting codeine is crucial due to its widespread use as an opioid analgesic and its potential for abuse, environmental contamination, and health risks. Codeine is often excreted unmetabolized or as metabolites into wastewater, leading to its presence in surface water and potentially affecting aquatic ecosystems [101].

The detection of codeine using BDDEs was explored by Švorc et al. in a study that employed DPV as the primary method. The research demonstrated an oxidation peak at +1.0 V vs. Ag/AgCl in Britton–Robinson buffer (pH 7.0) with a detection limit of 0.08 µM. The study’s novelty lies in the application of unmodified BDDEs for codeine determination without the need for surface modification, achieving high sensitivity and selectivity in pharmaceutical tablets and human urine samples [102].

Freitas et al. focused on utilizing a BDDE in a flow-based system for codeine detection. The approach involved batch injection analysis (BIA) combined with DPV, where the unmodified BDDE displayed excellent performance with minimal electrode fouling. The authors highlighted a detection limit of 0.05 µM and emphasized the electrode’s stability over multiple analyses. This method’s novelty is its integration into a flow-based analytical system, providing a rapid and reliable alternative for monitoring codeine in biological matrices [103].

The detection of adrenaline in the environment is important due to its role as a potent bioactive compound that can influence both ecosystems and public health. Adrenaline is excreted by humans and animals and can enter aquatic systems through wastewater, particularly in areas with high pharmaceutical or hospital waste discharge. Once in the environment, it can affect aquatic organisms by interfering with their hormonal balance, stress responses, and metabolic processes [104].

Švorc et al. investigated the electrochemical detection of adrenaline using a boron-doped diamond film electrode (BDDFE) and SWV as the primary method. The study was conducted on human urine samples spiked with adrenaline, with the oxidation peak observed at +0.75 V vs. Ag/AgCl in 0.5 M perchloric acid (HClO_4_). The optimized SWV parameters yielded a detection limit of 0.21 µM and a linear range from 0.7 to 60 µM. The novelty of the study lies in the use of an unmodified BDDFE, which provided high sensitivity, excellent repeatability (RSD of 3.5%), and robust performance without the need for electrode modification or complex sample preparation [105].

Fisetin is a naturally occurring flavonoid found in various fruits and vegetables and is known for its antioxidant, anti-inflammatory, and potential anti-cancer properties. Monitoring fisetin is important due to its increasing use in dietary supplements and pharmaceuticals, which may lead to environmental contamination through wastewater discharge. Boron-doped diamond electrodes (BDDEs) can be effectively used for fisetin detection, offering high sensitivity, chemical stability, and resistance to fouling, making them ideal for monitoring fisetin in complex environmental and biological samples [106].

The research conducted by Allahverdiyeva et al. employed a BDDE to quantitatively detect fisetin (FST), a plant polyphenol known for its significant therapeutic properties. The study utilized SWV and CV as primary methods, achieving irreversible oxidation of FST in 0.1 M nitric acid, with distinct anodic peaks at +0.69 V and +1.13 V. The detection limit was reported to be 0.08 µg/mL (2.8 × 10^−7^ mol/L), demonstrating high sensitivity and linearity over a wide concentration range (0.5–20 µg/mL). The novelty of this research lies in the optimization of the BDDE surface through cathodic pretreatment, which significantly enhanced the oxidation current of FST compared to untreated or anodically pretreated electrodes [107].

Synephrine is a naturally occurring alkaloid commonly found in citrus fruits and widely used in dietary supplements for weight loss and energy enhancement. Monitoring synephrine is important due to its potential cardiovascular effects, including increased heart rate and blood pressure, which raise concerns about its safety in high doses or long-term use [108,109].

The study conducted by Háššo and colleagues focused on the electrochemical detection of synephrine, a stimulant commonly found in dietary supplements, using a BDDE. The authors utilized both DPV and SWV for quantitative analysis. Synephrine showed an irreversible oxidation peak at +1.45 V versus the silver pseudoreference electrode in 2 M HClO_4_. The key advancement in this research is the use of a commercially available, unmodified BDDE, which eliminated the need for complex surface modifications and offered high reproducibility, selectivity, and minimal background current [110].

Nicotine detection is crucial due to its prevalence in environmental samples, including tobacco products and e-cigarette liquids. In the study by Mateusz Kowalcze et al., BDDEs were employed for the voltammetric detection of nicotine in e-cigarette liquids using DPV. The method achieved a detection limit of 0.01 mg/L with a linear range of 0.03 to 0.40 mg/L in an ammonium chloride solution at pH 7.5. The novelty of this research lies in its application to complex matrices without requiring extensive sample preparation, enhancing its practicality for rapid nicotine determination in commercial products [111].

Švorc et al. developed a sensitive and selective method for nicotine determination using DPV on a bare BDDE in Britton–Robinson buffer at pH 8. The method yielded an oxidation peak at +1.45 V, with a detection limit of 0.3 µM and a linear range from 0.5 to 200 µM. Unlike previous studies, this approach required minimal sample pretreatment and demonstrated excellent repeatability (RSD of 2.1%) in both tobacco products and anti-smoking pharmaceuticals, making it a promising tool for nicotine monitoring in various industries [112].

The detection of dopamine is crucial due to its vital role as a neurotransmitter in the central nervous system, where imbalances are linked to neurological disorders such as Parkinson’s disease, schizophrenia, and depression. Monitoring dopamine in biological and clinical samples is essential for understanding these conditions and developing effective treatments.

In the study by Haichao Li et al., a BDDE modified with gold and nickel nanoparticles was used for dopamine detection. DPV was employed, with measurements conducted in phosphate buffer at pH 7.0, closely mimicking physiological conditions. The study achieved a detection limit of 0.015 µM, with the novelty being the application of an anodic polarization treatment to enhance sensitivity and selectivity, allowing for clear separation of dopamine from common interferents like ascorbic acid [113].

In another study, Lytvynenko et al. utilized a bare BDDE for dopamine detection in neuron cultivation media. The research emphasized the electrode’s high resistance to fouling and excellent reproducibility in complex biological matrices. Using DPV at +0.75 V vs. Ag/AgCl, a detection limit of 2 µM was achieved. The novelty of this work lies in its application in real-time monitoring of neurotransmitters in living neuron cultures, showcasing BDDEs’ potential for in vitro and in vivo electrochemical sensing of dopamine [114].

Table 3 summarizes the results of BDDE electrode applications for the detection of pharmaceutical compounds.

## 7. Conclusions

The use of BDDEs has emerged as a highly effective and reliable approach for the detection of various toxic chemicals, including heavy metals, pesticides, pharmaceuticals, and endocrine disruptors. BDDEs’ unique properties, such as a wide potential window, low background current, and high resistance to fouling, make them superior to conventional electrode materials in electrochemical sensing. These advantages allow for the precise detection of analytes even in complex matrices, including environmental samples, biological fluids, and pharmaceutical products. The high stability of BDDEs ensures consistent performance over repeated measurements, which is critical for long-term monitoring applications.

In environmental monitoring, BDDEs have proven particularly valuable for detecting persistent pollutants such as pesticides and pharmaceutical residues in water and soil. Their ability to operate effectively without extensive sample pretreatment simplifies analytical procedures, reducing both time and costs. The electrodes’ resistance to fouling further enhances their applicability in real-world conditions, where matrix interference is often a significant issue. In addition, BDDEs’ high sensitivity enables the detection of trace levels of pollutants, contributing to more accurate risk assessments of environmental contamination.

For pharmaceutical monitoring, BDDEs offer an efficient solution for detecting drugs like dopamine, codeine, and nicotine in biological samples. These applications are crucial for both clinical diagnostics and forensic investigations, where precise quantification is essential. The low detection limits achieved using BDDEs highlight their potential for detecting ultra-trace levels of bioactive compounds, ensuring better control over drug usage and potential abuse.

Moreover, the use of BDDEs in the detection of endocrine disruptors, such as bisphenol A and pesticides with hormonal effects, underscores their importance in safeguarding public health. These compounds pose significant risks due to their ability to interfere with hormonal balance, and their reliable detection is vital for regulatory compliance and environmental protection. The integration of BDDEs with advanced electrochemical techniques, such as differential pulse voltammetry and square-wave voltammetry, further enhances analytical capabilities.

Overall, the growing body of research on BDDEs demonstrates their versatility and robustness across various fields, including environmental science, pharmaceutical analysis, and clinical diagnostics. Their adaptability to different detection needs, combined with their superior electrochemical properties, positions BDDEs as a leading material for next-generation sensors. Despite their many advantages, BDD electrodes still exhibit certain limitations, such as limited electrocatalytic activity toward some analytes and reduced sensitivity for non-electroactive species. Future research could focus on overcoming these drawbacks by modifying the BDD surface with various conductive materials, such as metal nanoparticles, carbon nanostructures, or redox-active polymers. These modifications may enhance electron transfer kinetics and extend the application of BDD-based sensors to a broader spectrum of environmental and biological contaminants. Integrating such hybrid materials with BDDEs may lead to the development of next-generation electrochemical sensors with improved analytical performance.

With continued advancements in BDDE technology, it is expected that its application scope will expand, contributing to more efficient pollutant monitoring, improved public health safety, and better management of environmental risks. Therefore, BDDEs represent a critical tool in addressing contemporary challenges related to pollution, health, and safety. Their eco-friendly nature and operational efficiency offer significant potential for sustainable analytical practices. This makes BDDEs not only a solution for current needs but also a promising technology for future developments in electrochemical sensing.

## Figures and Tables

**Figure 1 sensors-25-02339-f001:**
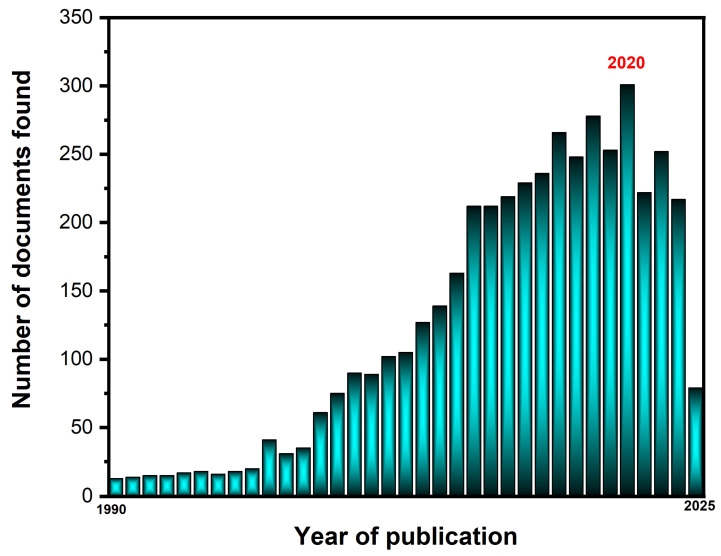
Annual number of publications retrieved from the Scopus database using the keyword “BDDE” (boron-doped diamond electrode) from 1990 to 2025.

**Figure 2 sensors-25-02339-f002:**
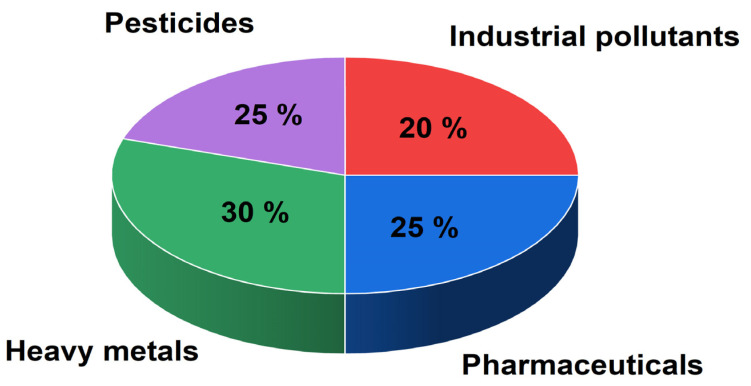
Percentage distribution of BDDE sensor applications across pollutant categories based on analysis of 100 scientific publications from 1990 to 2024 (Scopus and ScienceDirect databases). Analysis was based on a comprehensive search using targeted keyword combinations across pollutant types (e.g., “BDD electrode + heavy metals”, “BDD + pesticides”, etc.). The distribution is aligned with the major pollutant types discussed in Section 3, Section 4, Section 5 and Section 6 and reflects publication trends and focus areas in BDDE-based environmental sensing.

**Figure 3 sensors-25-02339-f003:**
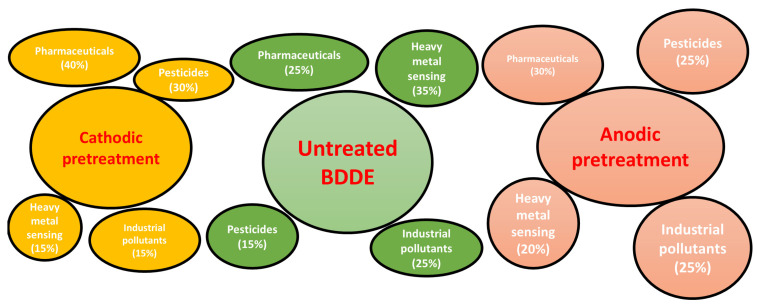
The effect of cathodic and anodic pretreatment on BDDEs compared to the untreated electrode presented as the percentage distribution of applications in detecting various pollutants. In total, 100 peer-reviewed articles published between 1995 and 2024 were included in this analysis. These were identified through keyword-based searches on Scopus, Google Scholar, and ResearchGate (e.g., “BDD electrode + pretreatment + pollutant type”), followed by manual screening to exclude duplicates and non-analytical studies.

**Table 3 sensors-25-02339-t003:** Overview of pharmaceuticals detected on BDDEs.

Analyte	pH	LOD (µM)	Technique	Type of Electrode Modifier	Linear Range (µM)	Ref.
Ciprofloxacin	5	0.05	SWV	/	0.15–2.11	[92]
	7	0.17	SWV	/	30–100	[93]
Acetaminophen	8	0.01	FIA	/	0.5–50	[83]
	7.4	0.05	DPNV	NSG	0.02–50	[84]
Paracetamol	3	0.16	DPV	CPT	0–13.87	[85]
	7	0.43	DPV	CNW	0.065–32	[86]
	7	0.065	DPV	/	0.05–50	[87]
Ibuprofen	/	5	DPV	APT	20–400	[88]
	2–5	0.41	DPV	/	0.949–66.9	[89]
Diclofenac	2	0.21	DPV	CPT	0.44–5.2	[98]
Codeine	7	0.08	DPV	/	0.1–60	[102]
Adrenaline	7	0.06	DPV	/	0.5–50	[103]
	6	0.21	SWV	/	0.7–60	[105]
Nicotine	7.5	0.0617	DPV	/	0.5–200	[111]
	8	0.3	DPV	/		[112]
Dopamine	7	0.015	DPV	Au/Ni NPs	8–10	[113]
	7	2	DPV	/	60–80	[114]

Abbreviations: CPT—cathodically pretreated; APT—anodically pretreated; NSG—nanometer-sized graphite; CNW—carbon nanowall.; SWV—Square wave voltammetry; Differential Pulse voltammetry; FIA—Flow injection analysis.

## Data Availability

Not applicable.

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
