# Peer review of "Boron-Doped Diamond Electrodes for Toxins Sensing in Environmental Samples—A Review"

_sensors, 2025, doi:10.3390/s25072339_

Round 1
Reviewer 1 Report
Comments and Suggestions for Authors
The paper entitled “Boron doped diamond electrode for toxins sensing in environmental samples - A review” by Mijajlovic et al. summarizes the recent development of the boron dope diamond (BDD) electrode for the detection of hazardous substances in the environmental samples. Several questions still can be addressed as follows:
- Paragraph 3, lines 71-72. Perhaps it would be better to elaborate on the advancement of electrode materials for use in electrochemical sensors first. What types of surface treatments are commonly employed to modify electrode modifier materials?
- Paragraph 4, lines 77-78. It consists of only one sentence and needs further elaboration or could be combined with the previous paragraph.
- Lines 80-81 need further elaboration about BDD thin films.
- Figure 1 needs to be inserted with the y-axis with the number of published papers
- In the introduction section, the authors should emphasize the advantages of using BDD electrodes over other types of carbon-based or metal-based electrodes in the development of electrochemical sensors.
- The utilization of BDD electrodes for detecting various types of toxic substances in lines 87-88 requires citation of the relevant literature. If possible, please elaborate
- It may be better to display the significant results (such as illustrations, graphs, or bar charts) from the selected paper cited in this manuscript, with permission from the publishers.
- Figure 2: how many papers have been used to make this percentage distribution? This figure could be made more informative if it were broken down into different types of toxin detection in the environment, as discussed in sections 3-6.
- The authors are suggested to draw a graphical abstract for the scope illustration of this manuscript
- Table 1 should mention the type of electrode modifier for metal ion determination and the number of paper citations still can be improved
- Table 2 can be created to summarize the analytical information for industrial pollutants detection like Table 1
- Figure 3: How to calculate the percentages of detecting various pollutants using BDD electrodes? How many papers are included in this analysis? The figure can still be made more informative.
- Lines 499-500, This sentence needs to be elaborated further
- Table 2 (page 15, line 561) should mention the type of electrode modifier for metal ion determination.
- Table 3 (page 22, lines 836-837) should mention the type of electrode modifier for metal ion determination.
- The authors are suggested to discuss the limitations of BDD electrodes for sensing purposes and the possibility of modifying the surface of BDD electrodes with different types of conductive materials to enhance its sensitivity
Reviewer 2 Report
Comments and Suggestions for Authors
The review fairly fully covers the use of boron dope diamond electrodes in the analysis of various objects. But I have some comments.:
- In Figure 1, the number of articles in the diagram must be indicated either on the ordinate axis or on the columns of the diagram.
- 2. Abbreviations are introduced at the first mention in the text and then only the abbreviation is written. It is necessary to check the entire manuscript! For example, in line 637 (DV) 638 (SWV), the abbreviations were not introduced at the first mention of differential pulse voltammetry and square-wave voltammetry.
- 3. There is no explanation of DPASV and LSV Technique in Table 1.
- 4. In lines 212, 215, 243,247, it is necessary to write the ion charges in upper indices.
